# Chromatin Accessibility and Transcriptional Landscape during Inhibition of *Salmonella enterica* by *Lactobacillus reuteri* in IPEC-J2 Cells

**DOI:** 10.3390/cells12060968

**Published:** 2023-03-22

**Authors:** Weiyun Qin, Zhanshi Ren, Chao Xu, Ya-nan Cao, Ming-an Sun, Ruihua Huang, Wenbin Bao

**Affiliations:** 1College of Animal Science and Technology, Yangzhou University, Yangzhou 225009, China; 2Institute of Comparative Medicine, College of Veterinary Medicine, Yangzhou University, Yangzhou 225009, China; 3College of Animal Science and Technology, Nanjing Agricultural University, Nanjing 210095, China

**Keywords:** *Lactobacillus reuteri*, *Salmonella enterica*, pigs, bacteriostasis, chromatin accessibility, transcriptome

## Abstract

*Lactobacillus reuteri* is a probiotic with bacteriostatic effects, which can effectively inhibit the activity of pathogens. However, the molecular mechanism underlying the inhibition of pathogens by *L. reuteri* in intestinal cells remains unclear. Using the porcine intestinal cell line IPEC-J2 as a model, we combined RNA-seq and ATAC-seq methods to delineate the porcine genome-wide changes in biological processes and chromatin accessibility in IPEC-J2 cells stimulated by *Salmonella enterica* BNCC186354, as well as *L. reuteri* ATCC 53608. Overall, we found that many porcine transcripts were altered after *S. enterica* BNCC186354 treatment, while *L. reuteri* ATCC 53608 treatment partially restored this alteration, such as *salmonella* infection and PI3K/AKT and MAPK pathways. Combined analysis of these two datasets revealed that 26 genes with similar trends overlapped between gene expression and chromatin accessibility. In addition, we identified potential host functional transcription factors (TFs), such as GATA1, TAL1, TBP, RUNX1, Gmeb1, Gfi1b, RARA, and RXRG, in IPEC-J2 cells that might play a critical role and are targeted by *L. reuteri* ATCC 53608. Moreover, we verified that PI3K/AKT, MAPK, and apoptosis pathways are potentially regulated by *S. enterica* BNCC186354 but restored by *L. reuteri* ATCC 53608. The PI3K/AKT pathway was activated by *L. reuteri* ATCC 53608, thereby potentially inhibiting *S. enterica* BNCC186354 infection. In conclusion, our data provide new insights into the expression pattern of functional genes and the epigenetic alterations in IPEC-J2 cells underlying the bacteriostatic action of *L. reuteri* ATCC 53608.

## 1. Introduction

*Salmonella* is a common foodborne pathogen, which is distributed among most pork production chains worldwide and passes through the technological steps of livestock and poultry production [1,2]. The *Salmonella* genome contains several *Salmonella* pathogenicity islands (SPIs), clusters of genes that encode virulence factors involved in different stages of *Salmonella* pathogenicity [3]. SPIs have multiple virulence genes, such as *hilA* and *invA*; *hilA* is required for downstream gene transcription and epithelial cell invasion [4], and the *invA* gene of *Salmonella* species allows the bacteria to invade the host and initiate infection, thereby increasing the degree of pathogenicity of the isolates. The amplification of nucleotide sequences within the *hilA* and *invA* genes of *Salmonella* has been evaluated as a means of detecting invasive *Salmonella* serovars [5]. Researchers have assessed 1095 retail fresh meats (chicken, pork, veal, lamb, beef, and turkey) and found that *Salmonella* was present in 38.1% of the samples [6]. *Salmonella* causes at least 1.35 million cases of salmonellosis every year in the United States, according to the Centers for Disease Control, with about 420 of those cases ending in death. The prevalence of *Salmonella* infection is increasing worldwide, with increasing numbers of livestock- and poultry-related outbreaks. *Salmonella,* especially *Salmonella enterica* (*S. enterica*), is also one of the most common pathogens causing diarrhea in pigs, which seriously endangers the sustainable development of the pig industry. The rate of *S. enterica* infection among weaned and finishing pigs is high, and salmonellosis among pigs can cause reduced weight gain and even death [7]. Therefore, it is critical for the swine industry to implement strategies to reduce the transmission of and infection with *Salmonella*. In the past few decades, the development of resistance to widely used antibiotics has been observed in a variety of bacteria, requiring the use of alternative strategies to control pathogens [8].

*Lactobacillus reuteri* (*L. reuteri*) can colonize the intestines of animals as a probiotic, where it produces antimicrobial organic material and reuterin to inhibit the growth of *Salmonella*, *Escherichia coli* (*E. coli*), and so on, thereby inhibiting intestinal infections, regulating the host immune function, and enhancing the body’s immunity [9,10,11,12]. In our previous study, 16S rRNA gene sequencing was used to find that *L. reuteri* comprises a relatively high proportion of probiotics in pigs, accounting for an average of 1% of the total bacteria, and is one of the dominant microorganisms in the intestine [13]. *L. reuteri* ATCC 53,608 isolated from pig intestines also has a similar antibacterial effect [14] and has the potential as a probiotic bacterium to inhibit *Salmonella* in piglets. 

*L. reuteri* has been found to strongly regulate interleukin family members, such as IL-8, IL-1α, and IL-6, in human epithelial cell lines [15,16]. In addition, *L. reuteri* can inhibit colitis and reduce the expression of proinflammatory cytokines in IL-10 gene-deficient mice [17,18]. Zhang et al. have revealed that *L. reuteri* inhibits *E. coli K88* and *S. enterica* in human intestinal epithelial cells and that higher adhesion force of *L. reuteri* to intestinal epithelial cells denotes a stronger pathogen-inhibitory effect [19]. *L. reuteri* KUB-AC5 confers both direct and indirect inhibitory effects on *Salmonella enterica* Typhimurium (STM) in the inflamed gut [20]. These studies suggest the evident anti-inflammatory action of *L. reuteri*. Several hypothesized mechanisms regarding the regulation of pathogens by *L. reuteri* in vitro have been proposed and validated, including enhancement of the epithelial barrier function (e.g., induction of mucins and preservation of tight junctions), inhibition of pathogens via microbial interactions (e.g., competition for nutrients and binding sites, production of antimicrobials), and the regulation of immune responses (e.g., regulation of cytokine expression, phagocytes, and T cell function; modulating Th1-promoting capacity) [21]. However, understanding epithelial biology is complicated by the fact that this tissue is in close proximity to the gut microbiota. How intestinal epithelial cells interact with the microbiota and how this is regulated at the gene expression level are critical questions. Kazakevych et al. have revealed how a highly conserved chromatin-remodeling factor has a distinct role in anti-microbial defense [22]. Nonetheless, the mechanism of epigenetic regulation remains unclear, especially in pigs. 

Cis-regulatory elements (CREs) are regulatory sequences relatively near their target gene, which play a crucial role in diseases and the growth performance in pigs [23,24,25]; however, annotation and identification of CREs also remain limited in pigs. Chromatin must be in an open state and then allow regulatory molecules to bind with DNA to function, in order for genes to be normally expressed [26]. Transcription factors (TFs) can interact specifically with CREs of genes, but the premise is chromatin remodeling must be induced by TFs in the nuclear structure, which leads to varying degrees of chromatin accessibility [27,28]. In the past few years, the Assay for Transposase-Accessible Chromatin with next-generation sequencing (ATAC-seq), as a potential tool, has been demonstrated to facilitate the identification of accessible open chromatin regions in a genome-wide manner [29]. The chromatin accessibility alterations in the bacteriostatic process of *L. reuteri* have not been investigated yet. Therefore, in this study, we used ATAC-seq combined with RNA-seq to study the genome-wide chromatin accessibility pattern of IPEC-J2 cells under the inhibition of *S. enterica* by *L. reuteri,* in order to reveal the transcriptional changes during the infection and antibacterial process, as well as to identify candidate regulatory TFs and pivotal signaling pathways for the antibacterial process of *L. reuteri*. Our study provides a valuable resource for elucidating transcriptional regulation and further comprehending the antibacterial mechanism of *L. reuteri* caused by epigenetic regulation.

## 2. Materials and Methods

### 2.1. Bacterial Culture

*L. reuteri* ATCC 53,608 strain was purchased from the China Center of Industrial Culture Collection (CICC; Beijing, China), and *S. enterica* BNCC186354 strain was purchased from the BeNa Culture Collection (BNCC; Beijing, China). *L. reuteri* ATCC 53,608 was grown in de Man, Rogosa, and Sharpe (MRS) medium at 37 °C for 24 h, and a 2.5 L anaerobic culture bag was used for maintaining anaerobic conditions. *S. enterica* BNCC186354 was inoculated in Luria–Bertani (LB) medium with general incubation. For subsequent cell assays, bacterial cells were collected via centrifugation at 4000 r/min for 5 min and washed with phosphate-buffer saline (PBS) three times. Finally, Dulbecco’s modified eagle medium (DMEM) with 10% fetal bovine serum was used to resuspend bacterial cells, which were adjusted to the densities of 1 × 10^7^ CFU/mL for *S. enterica* BNCC186354 and 1 × 10^8^ CFU/mL for *L. reuteri* ATCC 53,608 (Figure 1). All pH values were adjusted to 6.0 using NaOH solution. For AKT activation, IPEC-J2 cells were pretreated with 4 μg/mL of the AKT activator SC79 for 0.5 h and then treated with bacteria. 

### 2.2. Zone of Inhibition Assay

A total of 100 μL of *S. enterica* BNCC186354 fermentation broth was evenly spread onto the LB plate. After the plate had dried, Oxford cups with a pore size of 6.00 ± 0.10 mm were placed onto it. Then, 200 μL of *L. reuteri* ATCC 53,608 fermentation broth was added into each Oxford cup, and the plates were transferred into a 37 °C incubator for 18 h. The bacteriostatic zone was observed, and its diameter was determined. The diameter values were compared with the Kirby–Bauer antibiotic testing standard values prescribed by the Clinical and Laboratory Standards Institute, in order to determine whether they were resistant, susceptible, or intermediate [30].

### 2.3. Cell Viability Assay

IPEC-J2 cells transfected with siRNAs were counted and seeded into 96-well plates at a density of 3 × 10^3^ cells/well and detected using a Cell Counting Kit-8 (Beyotime Biotechnology, Shanghai, China) at 24, 48, and 72 h of *L. reuteri* ATCC 53,608 inoculation. Then, 10 µL of CCK8 solution was added to each well and incubated in an incubator for 2 h, followed by absorbance detection at 450 nm using a microplate reader.

### 2.4. Sample Collection

Three groups of IPEC-J2 cells were used for ATAC-seq and RNA-seq profiling, where a total of 18 samples were divided into the CTL group (control), SC group (*S. enterica* BNCC186354), and LR + SC group (*L. reuteri* ATCC 53,608 and *S. enterica* BNCC186354). IPEC-J2 cells were seeded into a 25 cm^2^ cell culture flask at a density of 0.7 × 10^6^ cells/mL. Cells were maintained with DMEM supplemented with 10% fetal bovine serum in a cell incubator (37 °C, under 5% CO_2_). When the cells were grown to complete confluence, they were incubated for 4 h in DMEM with *S. enterica* BNCC186354 or extra *L. reuteri* ATCC 53,608 added, without antibiotics [31,32,33]. In the control group, only the medium was replaced. Three replicate samples of each group were collected for high-throughput analysis.

### 2.5. Assay of Adhesion and Entry Ability of Bacteria into IPEC-J2 Cells

Unadhered bacteria were washed with PBS five times, following which the cultured cells were digested with trypsin. Colony counting was conducted to determine the bacterial adhesion and entry, MRS agar was used to identify *L. reuteri* ATCC 53608, LB agar was used to identify *S. enterica* BNCC186354, and the adhesion and entry differences between *S. enterica* BNCC186354 and *L. reuteri* ATCC 53,608 were calculated.

### 2.6. Detection of Salmonella Virulence Factors hilA and invA via a qPCR Assay

Total RNA was extracted using the Trizol protocol. The reaction mixture for cDNA synthesis contained 2 µL of 5× qRT SuperMix II and 500 ng total RNA, to which RNase-free H_2_O was added to make up a final volume of 10 µL. The reaction was carried out at 25 °C for 10 min, 50 °C for 5 min, and 85 °C for 5 min, and the reaction mixture was stored at 4 °C. After reverse transcription, hilA and invA expression was detected using AceQ^®^ Universal SYBR qPCR Master Mix (Q511-02, Vazyme Biotech Co., Ltd, Nanjing, Jiangsu, China). 16S rRNA was used as the house keeping gene for normalization. Primer details are provided in Appendix A. 

### 2.7. RNA-Seq Library Construction 

RNA-seq was performed using the method described as follows. Briefly, the libraries were generated with a total of 3 µg of RNA per sample, following the protocols for the NEBNext^®^ UltraTM RNA Library Prep Kit for Illumina^®^ (NEB, Ipswich, MA, USA). We used an AMPure XP system (Beckman Coulter, Beverly, MA, USA) to purify the fragments, and those with a length between 250 and 300 bp were selected. Then, cDNA was digested with USER Enzyme (NEB, USA), and the PCR process was subsequently performed. The purification of PCR products and the clustering of the index-coded samples were performed using an Agilent Bioanalyzer 2100 system (Agilent Technologies, Santa Clara, CA, USA) and a TruSeq PE Cluster Kit v3-cBot-HS (Illumina, Hayward, CA, USA), respectively. Finally, RNA-seq was performed on an Illumina Hiseq platform, according to standardized procedures.

### 2.8. RNA-Seq Data Processing

We used in-house Perl scripts to preliminarily process raw reads. First of all, Trimmomatic was used to process data, including cutting adapters and trimming low-quality bases. Meanwhile, we calculated the Q20, Q30, and GC contents. All subsequent analyses were based on the high-quality data obtained after processing. The index of the reference genome (Sscrofa11.1) was built, and paired-end clean reads were aligned to the reference genome using Hisat2 (version 2.0.5); meanwhile, the read numbers mapped to each gene were counted using Htseq-count. Then, the FPKM of each gene was calculated, based on the length of the gene and read count mapped to this gene. Differential expression analysis was performed using the DESeq2 R package (1.16.1). DESeq2 provides statistical routines for determining differential expression in digital gene expression data using a model based on the negative binomial distribution. The resulting *p*-values were adjusted using Benjamini and Hochberg’s approach for controlling the false discovery rate. Genes with an adjusted *p*-value < 0.05 found by DESeq2 were assigned as differentially expressed.

### 2.9. Fuzzy C-Means Clustering

Gene patterns were clustered using the soft clustering approach of the fuzzy c-means (FCM) algorithm implemented in the “Mfuzz” package (version 2.48.0, R foundation) [34]. The FCM algorithm allows each data point to belong to multiple clusters with varying degrees of membership, providing more flexibility than hard clustering. The cluster heatmap was drawn using the “pheatmap” package (version 1.0.12, R foundation).

### 2.10. ATAC-Seq Library Construction

ATAC-seq was performed according to Buenrostro’s method [29]. Briefly, 5 × 10^4^ cells were collected and lysed with cold lysis buffer, and the supernatant was removed. The transposing reaction system was configured with the Tn5 transposase. The cell nucleus was suspended with the transposing reaction system, and the DNA was purified after incubating it at 37 °C for 30 min. Then, the PCR amplification reaction was performed. The final DNA libraries were run on an Illumina platform after the DNA was purified.

### 2.11. ATAC-Seq Data Analysis

Cutadapt software was initially used to filter raw reads, including removing adapters and low-quality reads [35]. Then, clean reads were provided in FASTQ format for bioinformatics analysis. Bowtie 2 software [36] was used to compare the high-quality reads obtained from the sequencing of each sample with the reference genome of Sus scrofa (Sscrofa11.1, INSDC Assembly: GCA_000003025.6). DeepTools (version 2.07) [37] was used to map the density distribution of sequencing read in the 3 kb around the TSS of each gene. The MACS2 (version 2.1.1) software [38] was used for peak calling, with a cut-off of FDR < 0.05, and ChIPseeker software [39] was used for peak annotation. The peak regions were annotated, in order to indicate whether the peak region was in the TSS, exon, 5′-UTR, 3′-UTR, or intronic or intergenic region. DiffBind software [40] was used for differential peak analysis with the following settings: fold-change ≥ 1.5, *p*-value ≤ 0.05.

### 2.12. Pathway Analysis

Pathway analysis was performed to explore the annotated genes from DAR- or DEG-enriched pathways. The annotated genes or differential genes were analyzed based on the Kyoto Encyclopedia of Genes and Genomes (KEGG) database, in order to obtain all of the involved pathway terms. The significance level of each pathway term was calculated by performing the Fisher test, and *p*-values < 0.05 were considered to indicate statistical significance. 

### 2.13. Integrative Analysis of ATAC-Seq and RNA-Seq Data

Integrative analysis of ATAC-Seq and RNA-Seq data was performed in order to identify the TFs that made a considerable contribution to the regulation, according to the chromatin opening region and regulatory mechanism of these TFs based on the related downstream genes. The overlap between the up-regulated genes and chromatin regions with increased accessibility was calculated, and Fisher’s Exact Test was applied to determine whether their overlap was significant. The overlap between the down-regulated genes and chromatin regions with decreased accessibility was determined and tested using the same procedure.

### 2.14. ChIP-qPCR

IPEC-J2 cells were fixed with 1% formaldehyde for 10 min, quenched with 2.5 M glycine for 5 min, and sonicated to fragments of 200–500 bp in length. Subsequently, the chromatin fragments were incubated with anti-H3K4me3 and anti-H3K27ac, respectively, and were reverse cross-linked. ChIP-DNA was purified for qPCR. Primer details are provided in Appendix A.

### 2.15. MEME Analysis

MEME-ChIP 4.11.2 was used to identify and annotate the motif [41], using Tomtom to align the detected motif sequence with a known motif.

### 2.16. Quantitative Real-Time PCR (qRT-PCR)

The total RNA was extracted from IPEC-J2 cell samples with the Trizol method. Complementary DNA (cDNA) was synthesized using a PrimeScript RT reagent kit (TaKaRa, Beijing, China). qPCR SYBR Green Master Mix (Vazyme, Nanjing, China) was used in the qRT-PCR reaction. *GAPDH* was used as a reference gene, and primer details are listed in Appendix A. The CT values were calculated using the comparative Ct (2−ΔΔCt) method.

### 2.17. Western Blot Assay

Cell proteins were extracted and quantified using a BCA protein assay kit (Beyotime Biotechnology, Shanghai, China), and 20 ug of protein was separated by 10% SDS-PAGE gels and transferred to 0.22 μm PVDF membranes (Millipore, MA, USA). The membranes were blocked with 5% skim milk powder and incubated with primary antibodies at 4 °C overnight. The membranes were then incubated with the corresponding secondary antibodies, and an ECL detection system (Bio-Rad, Hercules, CA, USA) was used to detect the protein bands. HSP90 was used as controls. The primary antibodies and secondary antibody used are described in Appendix A. All Western blot images were normalized, the phosphorylated proteins were divided by corresponding total protein, and other proteins were divided by HSP90; the control was equal to 1.

### 2.18. Cell Cycle and Apoptosis Assay

Cell cycle and apoptosis assays were performed according to the instructions of the Cell Cycle and Apoptosis Analysis Kit and the Annexin V-FITC Apoptosis Detection Kit (Solarbio, Beijing, China), respectively. Briefly, for cell cycle detection, cells were collected and fixed with 70% iced ethanol at 4 °C overnight and then incubated with 0.5 mL of propidium iodide staining solution at 37 °C for 30 min, followed by flow cytometry at 488 nm. For apoptosis detection, cells were also collected and counted. Then, 195 µL of Annexin V-FITC binding solution, 5 µL of Annexin V-FITC, and 10 µL of a propidium iodide staining solution were successively added to 1 × 10^5^ cells. This was mixed well and incubated at room temperature, protected from light, for 20 min, and loaded into the flow cytometer for detection. 

### 2.19. Statistical Analysis and Data Availability

All replicates in each group are presented as the mean ± SD. SPSS software was used for statistical analysis. A two-sided Student’s t-test was used to analyze the differences between two groups. In all analyses, * denotes *p* < 0.05, ** denotes *p* < 0.01, and *** denotes *p* < 0.001 for the comparisons of the indicated treatments.

## 3. Results

### 3.1. L. reuteri ATCC 53,608 Inhibits S. enterica BNCC186354 In Vitro

In order to investigate the antibacterial effect of *L. reuteri* ATCC 53,608, we conducted zone of inhibition and bacterial adhesion assays. The results indicated that the growth of *S. enterica* BNCC186354 was inhibited in vitro, with its inhibition zone diameter reaching 15.31 ± 0.50 mm, categorizing it as susceptible (Figure 1A). Next, the inhibitory effect of *L. reuteri* ATCC 53,608 on *S. enterica* BNCC186354 adhesion to IEPC-J2 cells was assessed via a bacterial adhesion assay, for which we first selected an appropriate treatment concentration based on a CCK8 assay. We chose the concentration of 10^8^/mL *L. reuteri* ATCC 53,608 to treat IPEC-J2 cells with no significant effect on cell viability (Figure 1B); this concentration has also been commonly used in previous studies [31,32,33]. The adhesion and entry of *S. enterica* BNCC186354 showed a significant decrease (*p* < 0.05) with additional *L. reuteri* ATCC 53,608 treatment (Figure 1C). The adhesion and entry of *L. reuteri* ATCC 53,608 in the *S. enterica* BNCC186354 group was significantly higher than that in the control group (*p* < 0.05) (Figure 1D). In addition, detection of the expression of *Salmonella* virulence factors *hilA* and *invA* showed that the addition of *L. reuteri* ATCC 53,608 could significantly inhibit the expression of these two genes (*p* < 0.05) (Figure 1E). However, analysis of phenotype changes did not reflect deeper mechanisms, and thus, we used a combined RNA-seq and ATAC-seq analysis to perform an in-depth analysis of the inhibition effect of *L. reuteri* ATCC 53,608 on *S. enterica* BNCC186354.

### 3.2. Transcriptional Landscape of IPEC-J2 Cells under the Inhibition of S. enterica BNCC186354 by L. reuteri ATCC 53608

We used RNA-seq combined with ATAC-seq to study the transcriptional landscape and genome-wide chromatin accessibility pattern of IPEC-J2 cells under the inhibition of *S. enterica* BNCC186354 by *L. reuteri* ATCC 53608. The sequencing and mapping statistics are detailed in Appendix A. Principal component analysis (PCA) showed that each group could be well-distinguished (Figure 2A). Quality analysis showed good gene expression and distribution for subsequent differential expression analysis (Appendix A). The differentially expressed genes (DEGs) were subsequently analyzed between each group, where there were 417 DEGs between the *S. enterica* BNCC186354 and control groups and 598 DEGs between the *S. enterica* BNCC186354 with and without *L. reuteri* ATCC 53,608 groups (Figure 2B,C; Appendix A). KEGG enrichment analysis showed that several immune-related pathways were significantly enriched after *S. enterica* BNCC186354 treatment, such as the MAPK pathway (Figure 2D, Appendix A), which has been reported to be involved in regulating the infection process of *S. enterica* [42]. Meanwhile, some pathways associated with diseases, such as Alzheimer, Huntington, and others, were enriched after additional treatment with *L. reuteri* ATCC 53,608 (Figure 2E, Appendix A). 

Further, we sought to identify the antibacterial effect of *L. reuteri* ATCC 53608, regarding transcriptional changes and broad expression patterns, through clustering. The calculated optimal cluster number (k = 4) was used (Appendix A), and hub genes (minimum membership values > 0.5; Appendix A) were subjected to the k-means clustering algorithm, based on their centered and scaled average expression values (Figure 3A). The expression levels of cluster 1 and 3 genes were down-regulated by *S. enterica* BNCC186354 and then restored by *L. reuteri* ATCC 53608, while the expression levels of cluster 4 genes were up-regulated by *S. enterica* BNCC186354 and then restored by *L. reuteri* ATCC 53,608 (Figure 3B,C). In order to explore the underlying reasons for these observations, we hypothesized that *L. reuteri* ATCC 53,608 acts as a bacteriostatic agent by restoring the normal expression pattern of cluster 1, 3, and 4 genes, which were activated or suppressed. Further enrichment analysis revealed that cluster 1, 3, and 4 genes were mainly clustered into related pathways, such as the cell cycle, pathways of neurodegeneration–multiple diseases, and MAPK pathway (Figure 3D). It is noteworthy that the MAPK pathway was enriched not only in cluster 3, but also in cluster2, indicating that the genes of this pathway were both up- and down-regulated after *S. enterica* BNCC186354 treatment, and the down-regulated genes could be partially recovered by *L. reuteri* ATCC 53608. Contrary to our expectation, *Salmonella* infection was also enriched in clusters 1 and 3 (Appendix A), and thus, we reviewed the DEGs and found that only six DEGs, such as *AHNAK*, *MAP2K1*, *MAPK10*, *ARL8B*, *IL6*, and *PFN2*, were enriched after *S. enterica* BNCC186354 infection (Appendix A), but only *ARL8B* and PFN2 were downregulated (Appendix A). Therefore, the widespread downregulation of the *Salmonella* infection pathway after *Salmonella* infection may not be significant. The expression levels of cluster 2 genes continued to increase, indicating some pathways associated with immunity (Appendix A). 

### 3.3. Genome-Wide Changes in Chromatin Accessibility Reveal a Pathogenic and Bacteriostatic Process 

We prepared ATAC-seq libraries from IPEC-J2 cells to identify the chromatin accessibility induced by *S. enterica* BNCC186354 or *L. reuteri* ATCC 53608. The sequencing and mapping statistics are provided in Appendix A. Based on the mapped reads, Principal Component Analysis was performed, which revealed that the *S. enterica* BNCC186354 treatment formed well-scribed groups, compared with the control group; meanwhile, no clear distinction was observed after treatment with *L. reuteri* ATCC 53,608 (Appendix A). We detected the fragment size distribution and, as expected, the fragment size distribution presented a periodic change pattern (Appendix A). The density distribution of the reads around transcription start sites (TSSs) and the proportion of peaks were calculated and met a criterion (Appendix A). H3K4me3 and H3K27ac are known to mark active promoters [43]. Subsequently, we randomly selected several differentially accessible regions (DARs) and performed Chip-qPCR to detect the enrichment of H3K4me3 and H3K27ac in their promoter regions and found that histone H3K4me3 and H3K27ac marks were differentially enriched in accessible chromatin regions (Appendix A). We explored the altered chromatin accessibility after different treatments, and thousands of differing peaks with their respective annotated genes were identified among the different groups (Appendix A). These genes were significantly enriched in the PI3K/AKT pathway, MAPK pathway, Rap1 pathway, and so on (Figure 4A–D; Appendix A), similar to the pathways enriched in RNA-seq. Integrative analysis of matrix data was performed from ATAC-seq and RNA-seq. Differential Circos whole-genome visualizations were plotted using the significant DEGs and DARs (Appendix A). Venn diagrams revealed the interacting genes between RNA-seq and ATAC-seq among the different treatment groups (Appendix A). In order to screen the key regulatory molecules in the antibacterial process of *L. reuteri* ATCC 53608, we intersected the DEGs and DARs at a *p*-value < 0.05 and screened a total of 135 genes (Figure 4E), where 26 of them presented similar trends between gene expression and chromatin accessibility (shown in Figure 4F, in the form of a heat map; and DENND6B and PSAT1, as randomly displayed genes, are visualized using integrative genomics viewer (IGV) in Figure 4G).

### 3.4. L. reuteri ATCC53608 Regulates the Apoptosis and PI3K/AKT Pathway of IPEC-J2 Cells Induced by S. enterica BNCC186354

Further, we performed motif enrichment analysis of the altered chromatin accessibility (Figure 5A,B), and six common motifs were intersected among the enriched motifs (Figure 5C). Among these, GATA1 and TAL1 promote stress erythropoiesis and down-regulate immune response genes [44], TBP is a key protein in the transcription initiation of eukaryotic cells [45], and Gmeb1 modulates glucocorticoid receptor transactivation and inhibits cell apoptosis [46]. Some of these TFs may play a central role in the *S. enterica* BNCC186354 infection process and/or the antibacterial response of *L. reuteri* ATCC 53,608. We then performed KEGG enrichment analysis of genes annotated by DARs that may harbor these six motifs, in which a pathway of interest—the PI3K/AKT pathway—was identified (Figure 5D, Appendix A). PI3K/AKT is a key regulatory pathway, which plays a crucial role in cell growth and angiogenesis [47,48,49], and PIK3R1 can regulate the PI3K/AKT pathway through negative feedback [50].

qRT-PCR and Western blot results indicated that PIK3R1 was up-regulated after *S. enterica* BNCC186354 infection, while *L. reuteri* ATCC 53,608 treatment inhibited its expression; meanwhile, the peaks that were annotated to PIK3R1 showed the same trend (Figure 6A–C). Subsequently, by detecting the phosphorylation levels of its downstream signaling molecules, it was found that *S. enterica* BNCC186354 inhibited the phosphorylation of AKT, while the addition of *L. reuteri* ATCC 53,608 partially restores the Akt phosphorylation level (Figure 6C). Conversely, the phosphorylation of JNK1 and ERK1 was facilitated by *S. enterica* BNCC186354, but inhibited by *L. reuteri* ATCC 53,608 (Figure 6D). Meanwhile, in the presence of *S. enterica* BNCC186354, CDK4 and Cyclin D1 expression decreased, and the percentage of the G1 phase significantly decreased, while the G2 phase significantly increased (*p* < 0.01) (Figure 6D,E), the BCL-2/BAX ratio decreased, and the apoptosis of IPEC-J2 cells was significantly up-regulated (*p* < 0.01; Figure 6D,F); however, the addition of *L. reuteri* ATCC 53,608 partially restored PI3K/AKT and the phosphorylation and expression levels of other signal transduction proteins triggered by *S. enterica* BNCC186354 and phenotypically, reduced the apoptotic rate of IPEC-J2 cells, but had little effect on the cell cycle (Figure 6C–F). 

SC79, a unique specific AKT activator, activates AKT in the cytosol [51]; the *S. enterica* BNCC186354 adherence ability and virulence factors were detected to identify the effect of the PI3K/AKT pathway on the inhibition of *S. enterica* BNCC186354. The results showed that the numbers of adherent *S. enterica* BNCC186354 (*p* < 0.05) and expression of *hilA* and *invA* decreased (*p* < 0.01) after SC79 treatment (Figure 7A,B), indicated that the PI3K/AKT pathway potentially inhibits *S. enterica* BNCC186354 infection. These results suggest that *S. enterica* BNCC186354 promotes cell death and disrupts the G2/G1 cell-cycle and regulates PI3K/AKT, as well as MAPK signaling, while *L. reuteri* ATCC 53,608 restores the *S. enterica* BNCC186354-induced signaling changes, except for those to the cell cycle (Figure 7C).

## 4. Discussion

There has been increasing evidence that *L. reuteri* can effectively inhibit pathogens both in vitro and in vivo. For example, Arqués et al. (2004) have reported that 8 AU/mL of reuterin produced by *L. reuteri* PRO 137 presented bactericidal effects against *Staph. aureus*, *E. coli* O157:H7, *Salmonella choleraesuis subsp*. *choleraesuis*, and so on [52]. Our in vitro results also fully demonstrated this point of view, as *L. reuteri* ATCC 53,608 effectively inhibited *S. enterica* BNCC186354. To date, some hypotheses have been proposed to explain this antibacterial mechanism, but there exists no adequate mechanistic explanation, and reports on the related transcriptional regulation and epigenetics are scarce. As a model for *S. enterica* infection, intestinal epithelial cells (IECs) are an essential component of a highly regulated communication network that senses microbial and environmental stimuli, which can facilitate the activation of an IEC-specific immune response, thus activating the adaptive immune system [53,54]. Therefore, in this study, IPEC-J2 cells were used as an in vitro infection model for *S. enterica*, and integrative RNA-seq and ATAC-seq analysis was conducted to reveal the pathogenic process of *S. enterica* BNCC186354, as well as to focus on analyzing the alterations in chromatin accessibility and the transcriptional landscape associated with the antibacterial process of *L. reuteri* ATCC 53608.

Post-weaning diarrhea caused by *S. enterica* remains a major problem in the industry, causing decreases in performance and the survival of weaned pigs and posing a potential threat in terms of human food-borne bacterial diseases [55]. Researchers challenged piglets with *Salmonella* and revealed that the innate immune system Toll-like receptor cascades, phagosome pathway, cytokine signaling pathway, and lysosome pathway were first significantly upregulated at 2 dpi and then attenuated at 7 dpi [56]. Dar et al. [57] revealed that most of the enriched pathways were MAPK, NOD-like receptor, TLR, and PPAR signaling pathways, etc., after challenging the chicken with *Salmonella typhimurium*. Our study revealed that *S. enterica* BNCC186354 triggers host cellular responses through the RAS, MAPK, and RAP1 pathways, among others. These pathways are involved in processes, including inflammation and apoptosis, and some have been reported as being involved in regulating the infection process of *S. enterica* [58]. Interestingly, we found that the expression trend of genes was restored by *L. reuteri* ATCC 53,608 after *S. enterica* BNCC186354 infection; that is, these genes were mostly down-regulated under *S. enterica* BNCC186354 treatment only, while the addition of *L. reuteri* ATCC 53,608 inhibited this trend. These genes were concentrated in *Salmonella* infection, the cell cycle, MAPK, and other pathways that have been reported to be associated with *Salmonella* infection [42,59]. We were puzzled by this expression trend of *Salmonella* infection pathway genes, but we found that only six differentially expressed genes, such as *AHNAK*, *MAP2K1*, *MAPK10*, *ARL8B*, *IL6*, and PFN2, were enriched after *S. enterica* BNCC186354 infection, but only *ARL8B* and PFN2 were downregulated, from which we speculate that this pathway was not extensively activated after *S. enterica* BNCC186354, which may be related to the time and concentration of infection. In this regard, we hope to further elaborate on the credibility of this gene expression trend and the changes that occur at the epigenetic level through ATAC-seq.

Integrative ATAC-seq and RNA-seq analysis provides a good analytical basis for the overall transcriptional landscape and chromatin accessibility of bacteria, and the enriched pathways indirectly represent the processes of pathogen infection and antibacterial activity. A total of eight candidate TFs, such as GATA1, TAL1, TBP, RUNX1, Gmeb1, Gfi1b, RARA, and RXRG, were identified via MEME analysis, and these TFs were associated with cell activities, such as the immune response, transcription initiation, and apoptosis [44,45,46]. Further gene annotation and enrichment analyses were used to focus on the PI3K/AKT pathway, which is one of the most important intracellular pathways, regulating cell growth, motility, survival, metabolism, and angiogenesis [47,48]. The inhibition of PI3K/AKT can lead to reduced cell proliferation and increased cell death [49]. Our results also showed that protein changes in PIK3R1 are consistent with transcriptional changes and that downstream phosphorylation levels of AKT present opposite changes, consistent with Kong et al.’s study showing that PIK3R1 can regulate the PI3K/AKT pathway through negative feedback [50]. In addition, our results indicated that the activity of the PI3K/AKT pathway potentially inhibits *S. enterica* BNCC186354 infection in IPEC-J2 cells; the PI3K/AKT pathway may be activated by *L. reuteri* ATCC 53,608, thereby inhibiting *S. enterica* BNCC186354 infection. Phosphorylation levels of JNK and ERK, important proteins in the MAPK pathway, could also be partially restored by *L. reuteri* ATCC 53,608 following *S. enterica* BNCC186354 alterations. Studies have shown that the regulation of apoptosis by pathogens depends on the type of host cell and the stage of infection [60,61]. We showed that *S. enterica* BNCC186354 infection caused cell cycle arrest and apoptosis in IPEC-J2 cells, whereas *L. reuteri* ATCC 53,608 restored cell apoptosis induced by *S. enterica* BNCC186354, but had little effect on the cell cycle; cell cycle and apoptosis marker proteins also supported this phenomenon. Probiotics have been shown to prevent cytokine-induced apoptosis by activating AKT [62], and the results of this study also support the contention that *L. reuteri* ATCC 53,608 may restore *S. enterica* BNCC186354-activated apoptosis. It is worth noting that the pathways of neurodegeneration–multiple diseases, such as Huntington disease and Alzheimer disease, showed different associated transcriptional changes during the process in which *L. reuteri* ATCC 53,608 inhibits *S. enterica* BNCC186354. A previous study showed that *L. reuteri* SL001 plays positive roles in adjusting the intestinal bacterial community structure of AD model mice [63]. Furthermore, integration of the bacterial and fungal data sets at 12 weeks of age identified negative correlations between the Huntington’s disease-associated fungal species and *L. reuteri* [64]. 6-Hydroxydopamine (6-OHDA) is a widely used neurotoxin that leads to PD pathogenesis; the relative abundance of *L. reuteri* probiotic species in feces of 6-OHDA-lesioned mice was significantly decreased compared with those of sham-operated mice [65]. The above studies suggested that *L. reuteri* may be associated with some neurodegenerative diseases, which may also be an important reason for the enrichment of these related signaling pathways in IPEC-J2 cells after the addition of *L. reuteri* ATCC 53,608 in this study. As for some other pathways associated with viral infection that were also enriched, such as hepatocellular carcinoma, Kaposi sarcoma-associated herpesvirus infection, Epstein-Barr virus infection, and human papillomavirus infection, we speculate that this is due to the fact that many of the genes enriched in these pathways overlap with genes in the PI3K/AKT, MAPK, cell cycle, and apoptosis pathways. 

In RNA-seq screening, *Salmonella* infection, cell cycle, MAPK, and other signaling pathways were found to be down-regulated by *S. enterica* BNCC186354 and recovered by *L. reuteri* ATCC 53608. Some studies have used a tandem mass spectrometry (TMT-MS) approach to reveal strong proteomic disease-related changes not observed at the RNA level in Alzheimer’s disease, where MAPK/metabolism modules at the RNA level presented performance opposite to that of the corresponding protein modules [66]; this is somewhat similar to the results of this study, and thus, we speculate that this may be due to compensatory regulation occurring at the RNA level. Certainly, it is regrettable that we had difficulty in validating all of the screened pathways, and there must be some unexpected results that may help to further reveal the functional mechanisms of *L. reuteri* ATCC 53,608 in regulating the *S. enterica* BNCC186354 invasion of host epithelial cells.

## 5. Conclusions

In summary, our results fully displayed the chromatin accessibility and transcriptional landscape of *S. enterica* BNCC186354 infection, as well as the bacteriostatic action of *L. reuteri* ATCC 53,608 in IPEC-J2 cells, in particular, through the identification of eight TFs and their potential target PIK3R1. *S. enterica* BNCC186354 may regulate PI3K/AKT, MAPK, cell cycle, and apoptosis pathways after invading the host, while *L. reuteri* ATCC 53,608 can restore these pathways and the apoptotic phenotype. This study provides mechanistic guidance for the use of *L. reuteri* ATCC 53,608 as a probiotic in livestock, poultry, and humans and further provides a theoretical basis for the development and utilization of *L. reuteri* ATCC 53,608 and its synergistic drugs. 

## Figures and Tables

**Figure 1 cells-12-00968-f001:**
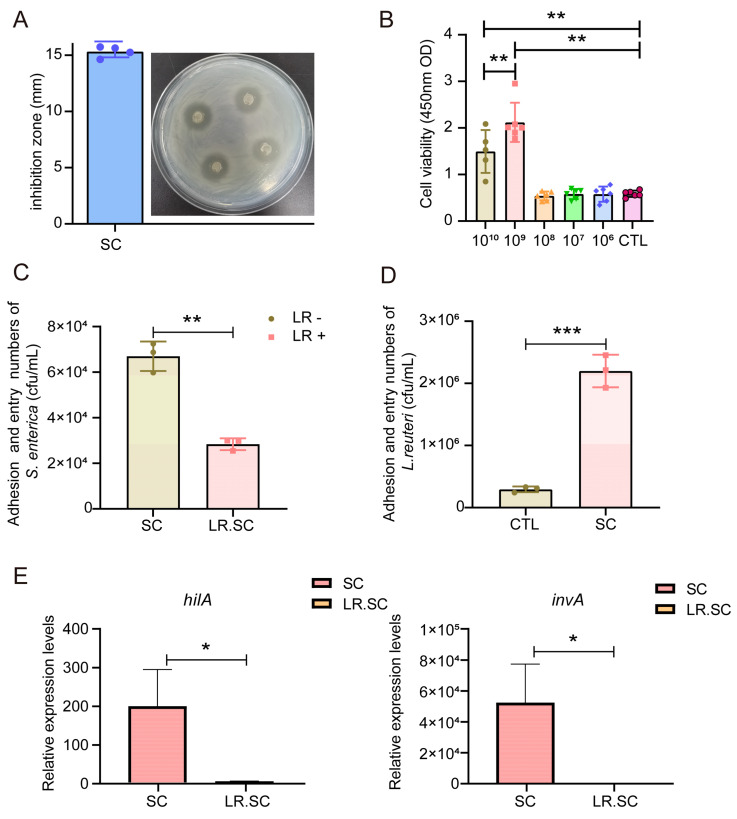
Antibacterial effect of *L. reuteri* ATCC 53,608 in vitro. (**A**). Zone diameter of inhibition of *L. reuteri* ATCC 53,608 against *S. enterica* BNCC186354, the blue dot represents the diameter of each inhibition zone (n = 4). (**B**). Effect of *L. reuteri* ATCC 53,608 treatment on the viability of IPEC-J2 cells, the different symbols represent the duplicated samples at different concentrations (n = 6). (**C**,**D**). Adhesion and entry of *S. enterica* BNCC186354 (**C**) and *L. reuteri* ATCC 53,608 (**D**) into IPEC-J2 cells between *S. enterica* BNCC186354 with and without *L. reuteri* ATCC 53,608 groups. (**E**). Expression of *hilA* and *invA* of *S. enterica* BNCC186354 between *S. enterica* BNCC186354 with and without *L. reuteri* ATCC 53,608 groups. CTL: control group, SC: *S. enterica* BNCC186354 treatment group, LR.SC: *L. reuteri* ATCC 53,608 and *S. enterica* BNCC186354 treatment group. LR−: without *L. reuteri* ATCC 53,608 treatment, LR+: with *L. reuteri* ATCC 53,608 treatment. *, *p* < 0.05; **, *p* < 0.01; ***, *p* < 0.001.

**Figure 2 cells-12-00968-f002:**
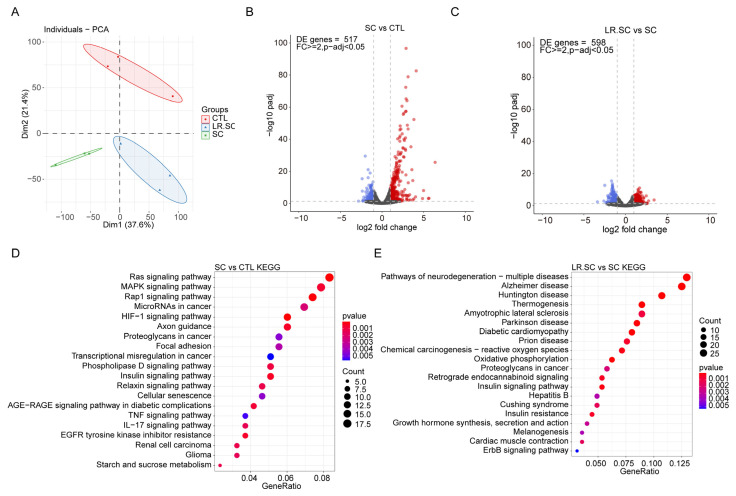
Quality metrics of RNA-seq data. (**A**). Principal component analysis (PCA) of all RNA-seq samples. (**B**,**D**). Volcano plot and top 20 signaling pathways of DEGs between *S. enterica* BNCC186354 and control groups. (**C**,**E**). Volcano plot and top 20 signaling pathways of all DEGs between *S. enterica* BNCC186354 groups with and without *L. reuteri* ATCC 53,608 treatment. The threshold was set as fold-change ≥ 2 and adjusted *p*-value < 0.05. CTL: control group, SC: *S. enterica* BNCC186354 treatment group, LR.SC: *L. reuteri* ATCC 53,608 and *S. enterica* BNCC186354 treatment group (the same below).

**Figure 3 cells-12-00968-f003:**
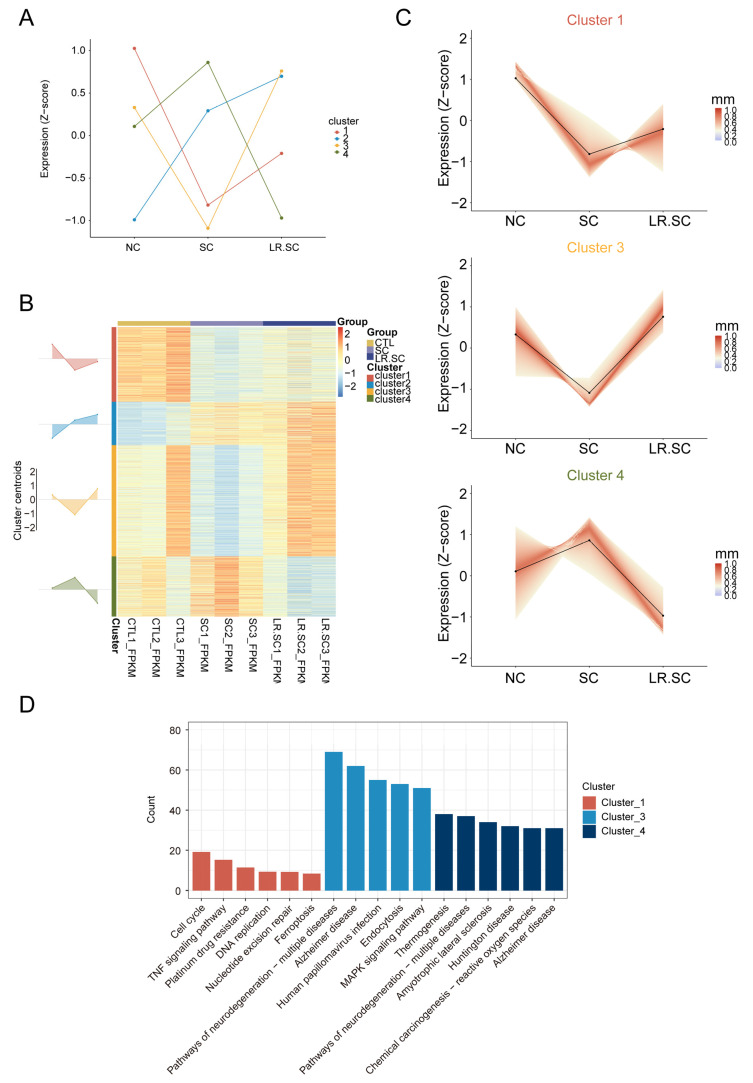
Different transcriptional changes during *L. reuteri* ATCC 53,608 treatment inhibit the *S. enterica* BNCC186354 process. (**A**). Line plot showing the dynamics of gene expression and cluster centroids identified (cluster number = 4) in different groups. (**B**). Heatmap representation of four defined clusters with differential gene expression dynamics in different groups. Area plots (left) show the overall gene expression dynamics of the clusters (visualized in relation to cluster centroids). (**C**). Line plots showing the dynamics of all genes (expression Z-score) within clusters 1, 3, and 4. Centroids are represented with black lines. The color density shows the correlation of a given gene with its centroid. (**D**). Gene enrichment analysis of the genes within clusters 1, 3, and 4; only the top five pathways are shown in each cluster.

**Figure 4 cells-12-00968-f004:**
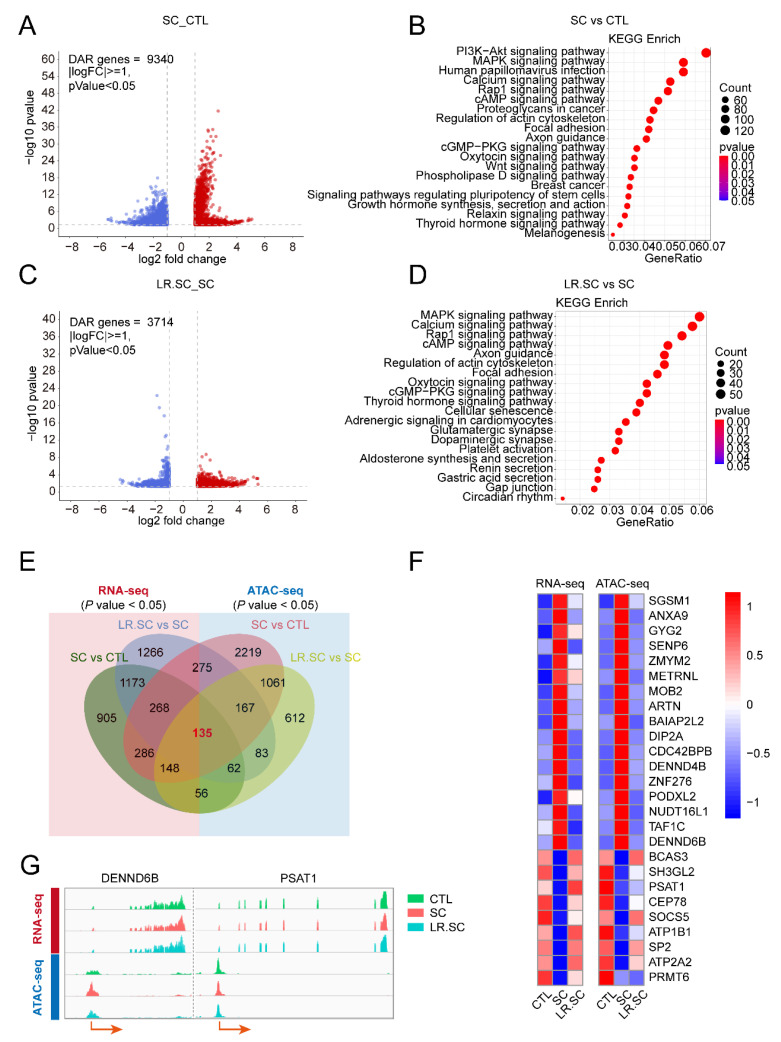
Genome-wide changes in chromatin accessibility induced by *S. enterica* BNCC186354 and *L. reuteri* ATCC 53608. (**A**,**B**). Volcano plot and top 20 signaling pathways of DEGs between *S. enterica* BNCC186354 and control groups. (**C**,**D**). Volcano plot and top 20 signaling pathways of DEGs between *S. enterica* BNCC186354 groups with and without *L. reuteri* ATCC 53,608 treatment. The threshold was set as a fold-change ≥ 1 and adjusted *p*-value < 0.05. (**E**). Venn diagram shows common DEGs and genes annotated by DARs between RNA-seq and ATAC-seq with a *p*-value < 0.05. The red background represents RNA-seq and the blue background represents ATAC-seq; the numbers marked in red represent the genes jointly screened among different comparison groups in ATAC-seq and RNA-seq data. (**F**). Heatmap of 26 genes with similar trends between gene expression and chromatin accessibility. (**G**) IGV was used to visualize expression and chromatin accessibility of DENND6B and PSAT1 genes.

**Figure 5 cells-12-00968-f005:**
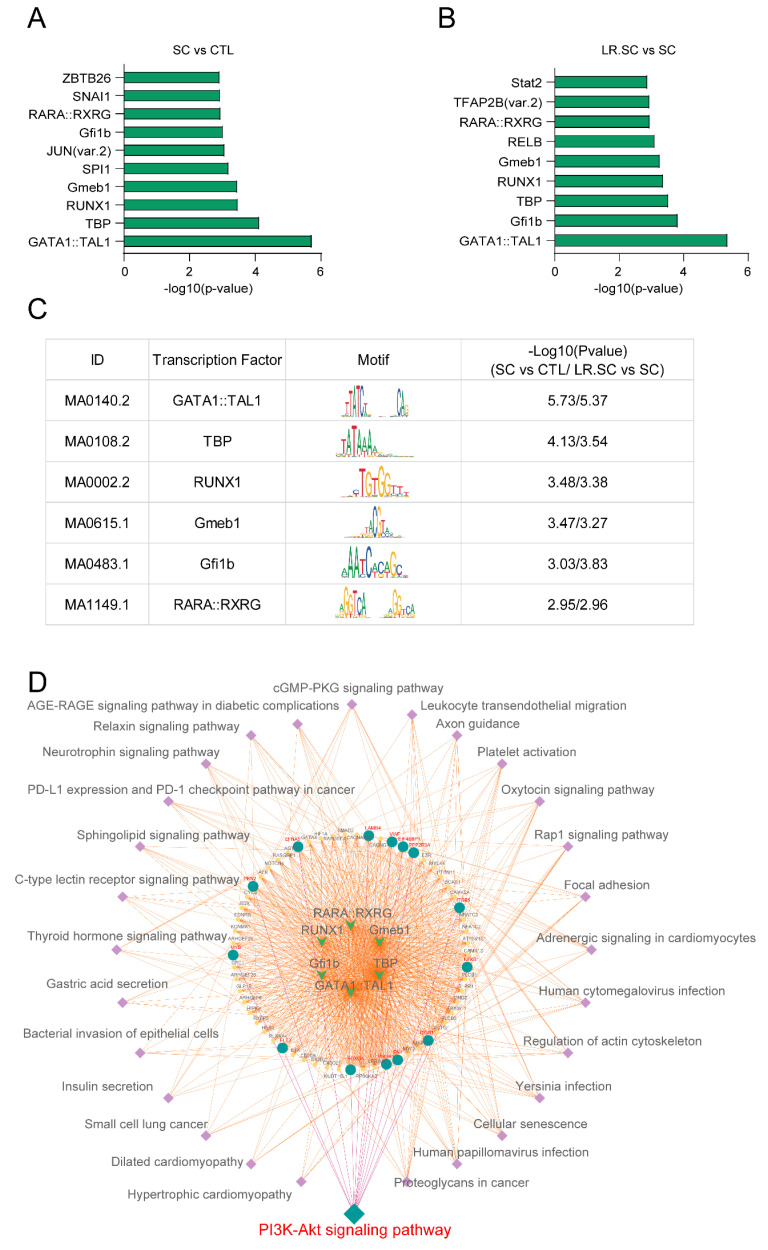
MEME analysis to identify functional TFs after different treatments in IPEC-J2 cells. (**A**,**B**). Top 10 transcription factor-binding motifs enriched in significantly increased and decreased peak regions, according to the *p*-values among different combinations of groups. (**C**). Common TFs and their corresponding motif details among different combinations of groups. (**D**) PPI interaction network of common TFs and genes annotated by TFs were subjected to KEGG analysis; the innermost inverted arrow shows TFs, the middle circle shows genes, the lateral square shows enriched pathways, and green circles are genes in the PI3K/AKT signaling pathway.

**Figure 6 cells-12-00968-f006:**
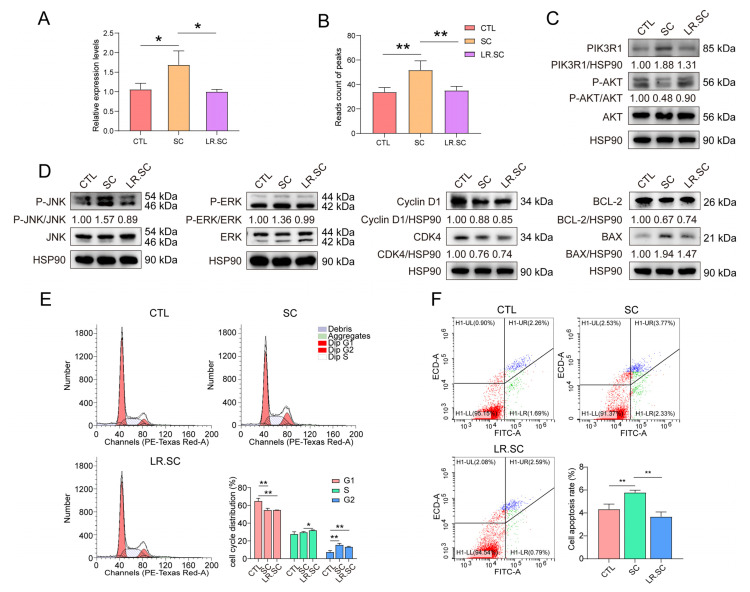
Detection of PI3K/AKT pathway, cell cycle, and apoptosis in IPEC-J2 cells treated with *S. enterica* BNCC186354 and *L. reuteri* ATCC 53,608 for 4 h. (**A**). PIK3R1 expression was detected via qRT-PCR among different groups. (**B**). Read counts of PIK3R1 peaks among different groups from the ATAC-seq dataset. (**C**). PIK3R1 protein expression and the phosphorylation level of AKT, detected by performing Western blotting. (**D**). Western blotting was performed to detect the phosphorylation level of JNK1 and ERK1 and the expression of Cyclin D1, CDK4, BCL-2, and BAX. All Western blot images were normalized, the phosphorylated proteins were divided by corresponding total protein, and other proteins were divided by HSP90; the control was equal to 1. (**E**). Distribution of cell cycle in IPEC-J2 cells treated with *S. enterica* BNCC186354 and *L. reuteri* ATCC 53608. (**F**). Effect of *S. enterica* BNCC186354 and *L. reuteri* ATCC 53,608 treatment on apoptosis in IPEC-J2 cells. *, *p* < 0.05; **, *p* < 0.01.

**Figure 7 cells-12-00968-f007:**
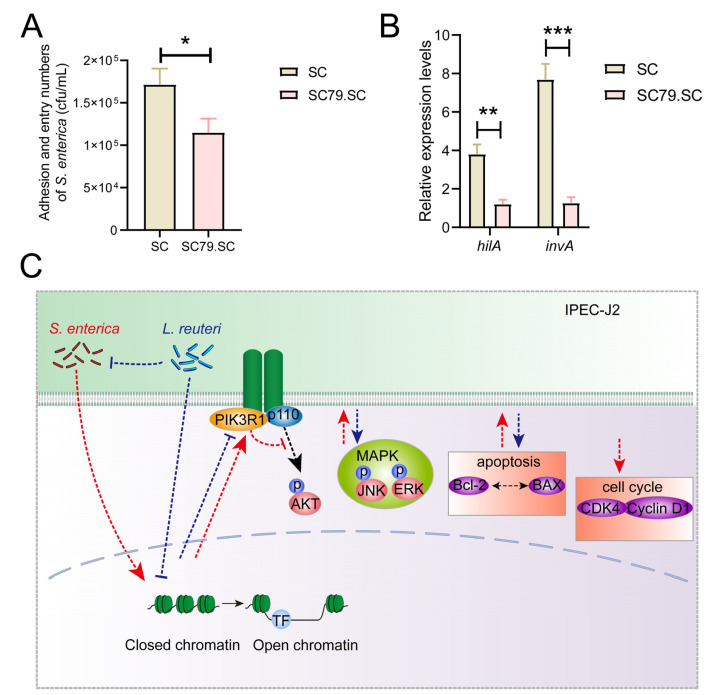
Activation of PI3K/AKT pathway in IPEC-J2 cells inhibits *S. enterica* BNCC186354 infection. **(A**,**B**) IPEC-J2 cells were pretreated with 4 μg/mL of the AKT activator SC79 for 0.5 h. Adhesion and entry numbers of *S. enterica* BNCC186354 (**A**) and expression of *hilA* and *invA* were detected (**B**). (**C**) Mechanistic hypothesis diagram for the changes in PI3K/AKT, MAPK pathways, cell cycle, and apoptosis in IPEC-J2 cells treated with *S. enterica* BNCC186354 and *L. reuteri* ATCC 53608. Red dashed lines represent the effect of *S. enterica* BNCC186354; blue dashed lines and crosses represent the effect of *L. reuteri* ATCC 53608. *, *p* < 0.05; **, *p* < 0.01; ***, *p* < 0.001. SC79.SC: SC79 and *S. enterica* BNCC186354 treatment group.

## Data Availability

All raw data of this study have been submitted to the NCBI SRA database (accessed on 21 March 2023 under accession number: PRJNA767941).

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
