# Peer review of "Chromatin Accessibility and Transcriptional Landscape during Inhibition of *Salmonella enterica* by *Lactobacillus reuteri* in IPEC-J2 Cells"

_cells, 2023, doi:10.3390/cells12060968_

Round 1
Reviewer 1 Report
The manuscript by Qin et al., depicts the effects of S. enterica and L. reuteri on porcine IPEC-J2 cells. They try to elucidate the antibacterial mechanisms of L. reuteri against S. enterica in this cell-culture infectious model.
Overall, the study and the figures are accurate and provide in addition to known effects such as growth and adherence inhibition, interesting observations regarding the chromatin accessibility, transcriptional patterns and their possible involment in metabolic pathways. They propose the involvement of epigenetic regulation in the bacteriostatic effect of L. reuteri.
This are valuable results and they are supported by the experimental results.
An aspect which is not clear enough motivated for me, is the infectious ration of approx. 10:1 between L. reuteri:S. enterica. Would the results look the same if you will use a 1:1 ratio? Why did you chose this ratio (excepting the fact that others did it)? Which ratios are to be expected in vivo? Can you please introduce/discuss such aspects through the manuscript?
It seems that beside modulation of infectious pathways, L. reuteri also influence the pathways of neurodegeneration. While for the pigs infectious treatments with L. reuteri will be unproblematic, could this treatment in humans contribute to neurodegenerative effects? Could you please introduce and discuss such aspects after a literature check?
Author Response
The manuscript by Qin et al., depicts the effects of S. enterica and L. reuteri on porcine IPEC-J2 cells. They try to elucidate the antibacterial mechanisms of L. reuteri against S. enterica in this cell-culture infectious model.
Overall, the study and the figures are accurate and provide in addition to known effects such as growth and adherence inhibition, interesting observations regarding the chromatin accessibility, transcriptional patterns and their possible involment in metabolic pathways. They propose the involvement of epigenetic regulation in the bacteriostatic effect of L. reuteri.
This are valuable results and they are supported by the experimental results.
An aspect which is not clear enough motivated for me, is the infectious ration of approx. 10:1 between L. reuteri:S. enterica. Would the results look the same if you will use a 1:1 ratio? Why did you chose this ratio (excepting the fact that others did it)? Which ratios are to be expected in vivo? Can you please introduce/discuss such aspects through the manuscript?
Reply:
Thanks for your comment, generally speaking, in the intestinal flora of healthy organisms, there are more probiotics and less pathogenic bacteria. In our previous study, 16S rRNA gene sequencing was used to find that L. reuteri is a relatively high proportion of probiotics in pigs, accounting for an average of 1% of the total bacteria, which is one of the dominant microorganisms in the intestine (PMID: 36288140), and the proportion of pathogenic bacteria is generally very low. Therefore, combined with the proportion used in others' research, we finally choose the infection ratio of 10:1. We refreshed the introduce in the revised manuscript.
It seems that beside modulation of infectious pathways, L. reuteri also influence the pathways of neurodegeneration. While for the pigs infectious treatments with L. reuteri will be unproblematic, could this treatment in humans contribute to neurodegenerative effects? Could you please introduce and discuss such aspects after a literature check?
Reply:
Thanks for your insightful comment. Previous study showed that L. reuteri SL001 play positive roles in adjusting the intesti-nal bacterial community structure of AD model mice (PMID: 33164464). Furthermore, integration of the bacterial and fungal data sets at 12 weeks of age identified negative correlations be-tween the Huntington's disease-associated fungal species and L. reuteri (PMID: 35262396). 6-Hydroxydopamine (6-OHDA) is a widely used neurotoxin that leads to PD patho-genesis, the relative abundance of L. reuteri probiotic species in feces of 6-OHDA-lesioned mice was significantly decreased compared with those of sham-operated mice (PMID: 31404072). The above studies suggested that L. reuteri may be associated with some neurodegenerative diseases, which may also be an important reason for the enrichment of these related signaling pathways in IPEC-J2 cells after the addition of L. reuteri ATCC 53608 in this study. We refreshed these references in the discussion.
Reviewer 2 Report
Dear Authors,
This study utilised a lots of experiments for unveiling the molecular mechanism underlying the inhibition of pathogene Salmonella enterica by Lactobacillus reuteri in porcine intestinal epithelial cells. The novelty of this paper mainly resides in ATAC-seq analysis for the chromatin assessibility and the subsequent apoptosis and PI3K/AKT pathway. The experimental works are tremendous, but the study design and the final mechanism hypothesis are not very clear because S. enterica and L. reuteri were simultaneously added to the IPEC-J2 host cells. In this way, the interactions can be between S. enterica and L. reuteri, S. enterica and IPEC-J2 cells, and L. reuteri and IPEC-J2 cells. The sebsequent inhibitory effects on the mentioned results in all the experiments can not only derive from the inhibition of L. reuteri on the invading S. enterica in different steps (into nucleus to affect chromatin, PIK3R1/AKT, MAPKJNKERK, apoptosis, and cell cycle) but also possibly be due to direct suppresison of S. enterica outside the host cells. This is my major concerns. In addition, the authors should address the influence of probiotic bacteria upon the chromatin accessibility of intestinal epithelial cells (e.g. Kazakevych J. Genome Biol. 2020;21(1):64. doi: 10.1186/s13059-020-01976-7) in the section of background for highlightening the novelty of this study. Compared with the adundant results of as many as 17 assays/experiments, the discussion is too short to reveal the depth of the study. Some of the results are not very reasonably involved in the inhibition of lactobacilli on Salmonella in epithelial cells, such as neurogeneration, Alzheimer disease, Huntington disease, cancers (HCC, Kaposi sarcoma, EBV infection, HPV infection, etc). However, neither explanations/comparisons with the literature nor hypotheses were proposed. The above concerns should be erased and its discussion is suggested to be fully strengthened.
For other concerns/errors in the details, please see the following list as below:
Line 3: Title
Salmonella Enterica and Lactobacillus Reuteri should be typed in italic as Salmonella enterica and Lactobacillus reuteri.
Line 18:
as model: as a model
Line 19:
combine: combined
Line 43
The prevalence of Salmonella is increasing worldwide: The prevalence of Salmonella infection is increasing worldwide
Line 55
Coli: coli
Line 75-76
The pattern of epigenetic regulation remains poorly annotated, especially in pigs
Line 116-117
Resistive: resistant
Line 141-148
2.6 Why were HILA and INV1 expressions detected? What is the rationale of quantifying these two virulence factors? No rationale was mentioned in the introduction.
Line 97-108:
Please insert Figure 1A in the appropriate site within the text of 2.1. Bacterial culture.
Line 275 Figure 1A. Workflow of L. reuteri ATCC 53608 and S. enterica BNCC186354 treatment.
Please describe precisely for its manipulation accordingly to which section of Materials and Methods.
Line 263-264; Line 275 Figure 1C.
What are the differences in their pH values (or medium colors) in different concentrations (10^6, 10^7, 10^8, 10^9, and 10^10/mL) of L. reuteri ATCC 53608?
Line 265-267; Line 275 Figure 1D and Figure 1E.
Are the pH values adjusted in the media with and without additional L. reuteri ATCC 53608 to normalize the effect of acidity in the media? In another word, how did the experiment convince readers the significant decrease in S. enterica is caused by L. reuteri rather than acidity of the medium?
Line 275 Figure 1D; Line 135-140 2.5.
Are the bacterial concentrations of S. enterica in the SC bar and the LR.SC representing the sum of adhesion and entry number of S. enterica?
Typing error of unit of Y-axis: Adhesio should be Adhesion
Line 275 Figure 1E; Line 135-140 2.5.
The same question to the Figure 1D. In addition, why was the SC bar (no treatment of LR) showing noticeable bacterial concentrations of L. reuteri?
Line 275 Figure 1F; Line 141-148. 2.6.
What is the house keeping gene used for normalization in the qRT-PCR for the mRNA expression levels of Salmonella virulence factors HILA and INV1? The unit of Y axis should be “relative expression level to *** gene (housekeeping gene)”.
Line 160-173 2.8. Line 181-186 2.10. Line 285-287
Line 300 Figure 2A.
Please insert Figure 2A in the appropriate sites within the text of 2.8. and 2.10.
The n = 18 in Figure 2A seems different from the description in 2.8. in which n = 15 divided into three groups. Please ensure which one is correct.
Were RNA-seq and ATAC-seq done from 9 independent in vitro treatment, respectively? Were RNA-seq and ATAC-seq done individually 9 times for each? Or were pooled RNA samples used for RNA-seq and ATAC-seq once? Judging from Figure 2B and Materials and Methods 2.4., three replicate samples of each group were collected for adhesion assay and high-throughput analysis. Please revise the n = 18 and n = 9 in Figure 2A accordingly for avoid misunderstanding.
Line 298-299
Alzheimer, Huntington and et . al were enriched after additional treatment with L. reuteri ATCC 53608.
Is there any reasonable explanation for such an association that looks like far from each other?
Line 311-313, Line 319, Line 494-496
The expression levels of cluster 1 and 3 genes were down-regulated by S. enterica and then restored by L. reuteri… related pathways, such as salmonella infection…
This result appeared not reasonable that “salmonella infection” pathway should be down-regulated by Salmonella but restored (up-regulated again) by Lactobacilli. Please describe more details of RNA-seq in this result within the “salmonella infection” pathway, and then discuss it with the details.
Line 325 Figure 3.
Pathways of neurodegeneration – multiple diseases, Huntington disease, Alzheimer disease showed associated transcriptional different changes during L. reuteri inhibits S. enterica process. Some other pathways such as hepatocellular carcinoma, Kaposi sarcoma-associated herpesvirus infection, EB virus infection, HPV infection are associated. What are the reasonable explanations for these associations?
Line 387 Figure 5, Line 474
MEME analysis: Please define it in the section and Materials and Methods.
Line 399
Figure 6B: should be Figure 6C
Line 400
Figure 6C: should be Figure 6D
Line 403
Figure 6C, D: should be Figure 6D, E
Line 404
Figure 6C, E: should be Figure 6D, F
Line 408
Figure 6C-E, E: should be Figure 6C-F?
Line 425-427
L. reuteri ATCC 53608 restores the S. enterica BNCC186354-induced signaling changes in phosphorylated signaling molecules except those related to the cell cycle.
Line 428 Figure 7A-B
In the results of Figure 7A-B, no LR.SC. group was studied for the restoration of S. enterica-induced signaling changes by SC79 treatment by L. reuteri.
In Figure C (Mechanism hypothesis diagram), the antibacterial effect of L. reuteri upon S. enterica outside the host IPEC-J2 cells should be delineated with a blue dashed line outside IPEC-J2 cells in Figure 7C, at least based on the result of Figure 1B (no presence of host cells). It would be more convincing to validate whether the bacteriocidal or bacteriostatic effect of L. reuteri upon S. enterica is present when both interact with each other with presence of host cells.
Another question regarding the direct effect of L. reutri upon IPEC-J2 cells has not been clearly answered. The blue dashed lines within host IPEC-J2 cells showed an indirect inhibitory effect of L. reuteri on the invading S. enterica, with the other blue dashed lines starting from the nucleus. However, whether the inhibitory (or preventive) effects of L. reuteri occurred directly upon host cells in different sites such as chromatin, PI3K/AKT, MAPK, apoptosis, cell cycle have not been clarified.
Line 474-475
A total of eight candidate TFs were identified by MEME analysis, and further gene annotation and enrichment analyses were used to focus on the PI3K/AKT pathway: This sentence is not clear. Please describe the eight TFs clearly for discussion.
Line 480
consistent with Kon et al.’s study: what is the exact finding of Kong et al.’s study? Please describe it concretely with citation.
Author Response
This study utilised a lots of experiments for unveiling the molecular mechanism underlying the inhibition of pathogene Salmonella enterica by Lactobacillus reuteri in porcine intestinal epithelial cells. The novelty of this paper mainly resides in ATAC-seq analysis for the chromatin assessibility and the subsequent apoptosis and PI3K/AKT pathway. The experimental works are tremendous, but the study design and the final mechanism hypothesis are not very clear because S. enterica and L. reuteri were simultaneously added to the IPEC-J2 host cells. In this way, the interactions can be between S. enterica and L. reuteri, S. enterica and IPEC-J2 cells, and L. reuteri and IPEC-J2 cells. The sebsequent inhibitory effects on the mentioned results in all the experiments can not only derive from the inhibition of L. reuteri on the invading S. enterica in different steps (into nucleus to affect chromatin, PIK3R1/AKT, MAPKJNKERK, apoptosis, and cell cycle) but also possibly be due to direct suppresison of S. enterica outside the host cells. This is my major concerns. In addition, the authors should address the influence of probiotic bacteria upon the chromatin accessibility of intestinal epithelial cells (e.g. Kazakevych J. Genome Biol. 2020;21(1):64. doi: 10.1186/s13059-020-01976-7) in the section of background for highlightening the novelty of this study. Compared with the adundant results of as many as 17 assays/experiments, the discussion is too short to reveal the depth of the study. Some of the results are not very reasonably involved in the inhibition of lactobacilli on Salmonella in epithelial cells, such as neurogeneration, Alzheimer disease, Huntington disease, cancers (HCC, Kaposi sarcoma, EBV infection, HPV infection, etc). However, neither explanations/comparisons with the literature nor hypotheses were proposed. The above concerns should be erased and its discussion is suggested to be fully strengthened.
Reply:
Thanks for your insightful comments. Regarding your major concerns about the regulatory mechanism and interactions among three factors, we agree with your viewpoint. Our study primarily addressed the transcriptional and epigenetic changes induced by Salmonella in IPEC-J2 cells and which signals were altered in IPEC-J2 cells following L. reuteri addition, but it is difficult to say whether L. reuteri occurred directly by suppressing Salmonella or by interacting with IPEC-J2 cells. However, it is noteworthy that we observed that the addition of L. reuteri in the absence of host cells directly inhibited Salmonella in vitro, while the addition of L. reuteri in the presence of host cells triggered alterations in some unique signaling pathways. Therefore, we speculate that the inhibition of L. reuteri is a very complex situation, and that it may both directly inhibit Salmonella and resist Salmonella invasion by altering signaling in IPEC-J2 cells. A perfect study is very difficult and our next step will also be to explore more deeply the mechanism by which L. reuteri resists Salmonella invasion of IPEC-J2 cells, but we are afraid that it will be difficult to achieve this in this study. Therefore, we weaken the hypothesis of our paper appropriately to better fit our experimental results. We hope that you will find our manuscript has been significantly improved after the revision. Please find the other detailed one-to-one response below.
For other concerns/errors in the details, please see the following list as below:
Line 3: Title
Salmonella Enterica and Lactobacillus Reuteri should be typed in italic as Salmonella enterica and Lactobacillus reuteri.
Reply:
Done.
Line 18:
as model: as a model
Reply:
Done.
Line 19:
combine: combined
Reply:
Done.
Line 43
The prevalence of Salmonella is increasing worldwide: The prevalence of Salmonella infection is increasing worldwide
Reply:
Done.
Line 55
Coli: coli
Reply:
Done.
Line 75-76
The pattern of epigenetic regulation remains poorly annotated, especially in pigs
Reply:
Done.
Line 116-117
Resistive: resistant
Reply:
Done.
Line 141-148
2.6 Why were HILA and INV1 expressions detected? What is the rationale of quantifying these two virulence factors? No rationale was mentioned in the introduction.
Reply:
The Salmonella genome contains several Salmonella pathogenicity islands (SPIs’), clusters of genes that encode virulence factors involved in different stages of Salmonella pathogenicity. SPI has multiple virulence genes such as HILA and INVA, the HILA is required for downstream gene transcription and epithelial cell invasion, the INVA gene of the Salmonella species allows the bacteria to invade the host and initiate infection, thereby increasing the degree of pathogenicity of the isolates, amplification of nucleotide sequences within the HLIA and INVA gene of Salmonella has been evaluated as a means of detecting invasive Salmonella serovars. We added this information in the introduction, and we performed qRT-PCR to quantifying these two virulence factors,
Line 97-108:
Please insert Figure 1A in the appropriate site within the text of 2.1. Bacterial culture.
Line 275 Figure 1A. Workflow of L. reuteri ATCC 53608 and S. enterica BNCC186354 treatment.
Please describe precisely for its manipulation accordingly to which section of Materials and Methods.
Reply:
Thanks for your comment, we think that inserting Figure1A into Materials and Methods will affect the overall aesthetics of the article, and will cause too many pictures of the paper to affect the readability, and the text of 2.1 is enough to illustrate the strategy we used, so we deleted Figure 1A.
Line 263-264; Line 275 Figure 1C.
What are the differences in their pH values (or medium colors) in different concentrations (10^6, 10^7, 10^8, 10^9, and 10^10/mL) of L. reuteri ATCC 53608?
Line 265-267; Line 275 Figure 1D and Figure 1E.
Are the pH values adjusted in the media with and without additional L. reuteri ATCC 53608 to normalize the effect of acidity in the media? In another word, how did the experiment convince readers the significant decrease in S. enterica is caused by L. reuteri rather than acidity of the medium?
Reply:
In this study, all pH values were adjusted to 6.0 using NaOH solution, and we added this detail in the 2.1 section.
Line 275 Figure 1D; Line 135-140 2.5.
Are the bacterial concentrations of S. enterica in the SC bar and the LR.SC representing the sum of adhesion and entry number of S. enterica?
Typing error of unit of Y-axis: Adhesio should be Adhesion
Reply:
Yes, the bacterial concentrations of S. enterica in different bars representing the sum of adhesion and entry number of S. enterica. We also revised the error of Y-axis.
Line 275 Figure 1E; Line 135-140 2.5.
The same question to the Figure 1D. In addition, why was the SC bar (no treatment of LR) showing noticeable bacterial concentrations of L. reuteri?
Reply:
Thanks for your suggestion, the answer to the first question is the same as above. We made a mistake about the x-axis, our labeling was wrong, and we have now revised the x-axis titles and data annotation in figure 1E.
Line 275 Figure 1F; Line 141-148. 2.6.
What is the house keeping gene used for normalization in the qRT-PCR for the mRNA expression levels of Salmonella virulence factors HILA and INV1? The unit of Y axis should be “relative expression level to *** gene (housekeeping gene)”.
Reply:
16S rRNA was used as the house keeping gene for ormalization in the qRT-PCR. We also revised the unit of Y axis.
Line 160-173 2.8. Line 181-186 2.10. Line 285-287
Line 300 Figure 2A.
Please insert Figure 2A in the appropriate sites within the text of 2.8. and 2.10.
The n = 18 in Figure 2A seems different from the description in 2.8. in which n = 15 divided into three groups. Please ensure which one is correct.
Were RNA-seq and ATAC-seq done from 9 independent in vitro treatment, respectively? Were RNA-seq and ATAC-seq done individually 9 times for each? Or were pooled RNA samples used for RNA-seq and ATAC-seq once? Judging from Figure 2B and Materials and Methods 2.4., three replicate samples of each group were collected for adhesion assay and high-throughput analysis. Please revise the n = 18 and n = 9 in Figure 2A accordingly for avoid misunderstanding.
Reply:
Thanks for your comment. However, we believe that inserting Figure 2A into the paper will affect the overall beauty of the article, and will lead to too many figures of the paper. As for the samples, n=18 is correct. RNA-seq and ATAC-seq were performed from 9 independent in vitro treatment, but RNA-Seq extracts RNA from cells. ATAC-seq identifies accessible DNA regions by probing open chromatin with hyperactive mutant Tn5 Transposase that inserts sequencing adapters into open regions of the genome. To avoid confusion, we changed the sample size in section 2.4 and removed Figure 2A. If there is a better suggestion, we are willing to listen to it.
Line 298-299
Alzheimer, Huntington and et. al were enriched after additional treatment with L. reuteri ATCC 53608.
Is there any reasonable explanation for such an association that looks like far from each other?
Reply:
Thanks for your insightful comment. Previous study showed that L. reuteri SL001 play positive roles in adjusting the intesti-nal bacterial community structure of AD model mice (PMID: 33164464). Furthermore, integration of the bacterial and fungal data sets at 12 weeks of age identified negative correlations be-tween the Huntington's disease-associated fungal species and L. reuteri (PMID: 35262396). 6-Hydroxydopamine (6-OHDA) is a widely used neurotoxin that leads to PD patho-genesis, the relative abundance of L. reuteri probiotic species in feces of 6-OHDA-lesioned mice was significantly decreased compared with those of sham-operated mice (PMID: 31404072). The above studies suggested that L. reuteri may be associated with some neurodegenerative diseases, which may also be an important reason for the enrichment of these related signaling pathways in IPEC-J2 cells after the addition of L. reuteri ATCC 53608 in this study.
Line 311-313, Line 319, Line 494-496
The expression levels of cluster 1 and 3 genes were down-regulated by S. enterica and then restored by L. reuteri… related pathways, such as salmonella infection…
This result appeared not reasonable that “salmonella infection” pathway should be down-regulated by Salmonella but restored (up-regulated again) by Lactobacilli. Please describe more details of RNA-seq in this result within the “salmonella infection” pathway, and then discuss it with the details.
Reply:
Thanks for your comment, we noticed that the salmonella infection was enriched in clusters 1 and 3, but we found that only 6 DEGs such as AHNAK, MAP2K1, MAPK10, ARL8B, IL6, and PFN2 were enriched after S. enterica BNCC186354 infection (Table S4), but only ARL8B and PFN2 were downregulated (Table S3). Therefore, the widespread downregulation of salmonella infection pathway after Salmonella infection may not be significant. For this trend of expression of salmonella infection pathway genes, we speculate that the pathway is not extensively activated after Salmonella infection, which may be related to the time and concentration of infection. Related modifications have been added to the paper.
Line 325 Figure 3.
Pathways of neurodegeneration – multiple diseases, Huntington disease, Alzheimer disease showed associated transcriptional different changes during L. reuteri inhibits S. enterica process. Some other pathways such as hepatocellular carcinoma, Kaposi sarcoma-associated herpesvirus infection, EB virus infection, HPV infection are associated. What are the reasonable explanations for these associations?
Reply:
The reasons for the enrichment of pathways related to neurodegenerative diseases have been described above, while other signaling pathways such as Epstein-Barr virus infection are enriched in cluster 1 with a p.adjust > 0.05, which is not statistically significant. Some other signaling pathways associated with viral infections such as Kaposi sarcoma - associated herpesvirus infection (https://www.kegg.jp/entry/map05167), EB virus get infection (https://www.kegg.jp/entry/map05169), HPV infection (https://www.kegg.jp/entry/map05165), many of the genes in these signaling pathways overlap with those in PI3K-Akt signaling pathway, MAPK signaling pathway, Cell cycle, Apoptosis and other pathways. This may also be an important reason why enrichment analysis can annotate pathways related to these viral infections.
Line 387 Figure 5, Line 474
MEME analysis: Please define it in the section and Materials and Methods.
Reply:
Done.
Line 399
Figure 6B: should be Figure 6C
Reply:
Done.
Line 400
Figure 6C: should be Figure 6D
Reply:
Done.
Line 403
Figure 6C, D: should be Figure 6D, E
Reply:
Done.
Line 404
Figure 6C, E: should be Figure 6D, F
Reply:
Done.
Line 408
Figure 6C-E, E: should be Figure 6C-F?
Reply:
Done.
Line 425-427
- reuteri ATCC 53608 restores the S. enterica BNCC186354-induced signaling changes in phosphorylated signaling molecules except those related to the cell cycle.
Reply:
Done.
Line 428 Figure 7A-B
In the results of Figure 7A-B, no LR.SC. group was studied for the restoration of S. enterica-induced signaling changes by SC79 treatment by L. reuteri.
Reply:
Thanks for your comment, we have shown in Figure 1 that the additional treatment of L. reuteri has a clear inhibitory effect on Salmonella. In this exprement, we aimed to explore whether the PI3K/AKT signaling pathway identified by sequencing could inhibit Salmonella infection. SC79 is a unique specific AKT activator, activates AKT in the cytosol [45]. So we used SC79 treatment to identify effect of PI3K/AKT pathway in the inhibition of S. enterica BNCC186354.
In Figure C (Mechanism hypothesis diagram), the antibacterial effect of L. reuteri upon S. enterica outside the host IPEC-J2 cells should be delineated with a blue dashed line outside IPEC-J2 cells in Figure 7C, at least based on the result of Figure 1B (no presence of host cells). It would be more convincing to validate whether the bacteriocidal or bacteriostatic effect of L. reuteri upon S. enterica is present when both interact with each other with presence of host cells.
Another question regarding the direct effect of L. reutri upon IPEC-J2 cells has not been clearly answered. The blue dashed lines within host IPEC-J2 cells showed an indirect inhibitory effect of L. reuteri on the invading S. enterica, with the other blue dashed lines starting from the nucleus. However, whether the inhibitory (or preventive) effects of L. reuteri occurred directly upon host cells in different sites such as chromatin, PI3K/AKT, MAPK, apoptosis, cell cycle have not been clarified.
Reply:
Thanks for your insightful comment, we did do the inhibition of Salmonella by L. reuteri without presence of host cells in Figure 1B, but in Figure 1D-F we did the inhibition of Salmonella by L. reuteri in the presence of IPEC-J2. It was shown that L. reuteri could inhibit Salmonella in the presence of host cells. But as the reviewer raised another question, whether the inhibitory (or preventive) effects of L. reuteri occurred directly upon host cells in different sites such as chromatin, PI3K/AKT, MAPK, apoptosis, cell cycle have not been clarified. Therefore, according to the reviewer's comments, the blue dotted line inside the cell was cancelled and the blue dotted line outside the cell was marked.
Line 474-475
A total of eight candidate TFs were identified by MEME analysis, and further gene annotation and enrichment analyses were used to focus on the PI3K/AKT pathway: This sentence is not clear. Please describe the eight TFs clearly for discussion.
Reply:
Thanks for your insightful comment, we described the eight TFs information.
Line 480
consistent with Kon et al.’s study: what is the exact finding of Kong et al.’s study? Please describe it concretely with citation.
Reply:
Kong et al.’s study showed that PIK3R1 can regulate the PI3K/AKT pathway through negative feedback, we added this information.
Round 2
Reviewer 2 Report
Thanks very much for the authors' quick revision. The followings are my concerns. Please also revise Figure 7C to make it more genuinely express the major results.
Line 141-148
2.6 Why were HILA and INV1 expressions detected? What is the rationale of quantifying these two virulence factors? No rationale was mentioned in the introduction.
Reply:
The Salmonella genome contains several Salmonella pathogenicity islands (SPIs’), clusters of genes that encode virulence factors involved in different stages of Salmonella pathogenicity. SPI has multiple virulence genes such as HILA and INVA, the HILA is required for downstream gene transcription and epithelial cell invasion, the INVA gene of the Salmonella species allows the bacteria to invade the host and initiate infection, thereby increasing the degree of pathogenicity of the isolates, amplification of nucleotide sequences within the HLIA and INVA gene of Salmonella has been evaluated as a means of detecting invasive Salmonella serovars. We added this information in the introduction, and we performed qRT-PCR to quantifying these two virulence factors,
Comment:
Line 40: SPIs’ should be SPIs.
Line 41-46: This sentence is too long and full of grammatical errors. Salmonella gene names should be expressed as hilA and invA in italics, or readers will get confused about the terms. Please also refer to your cited references [4] and [5]. All the HILA and INVA should be corrected into hilA and invA throughout the entire manuscript, including figures and tables.
SPI has multiple virulence genes such as HILA and INVA, the HILA is required for downstream gene transcription and epithelial cell invasion [4], the INVA gene of the Salmonella species allows the bacteria to invade the host and initiate infection, thereby increasing the degree of pathogenicity of the isolates, amplification of nucleotide sequences within the HLIA and INVA gene of Salmonella has been evaluated as a means of detecting invasive Salmonella serovars [5].
This sentence is suggested to be corrected as follow:
SPIs have multiple virulence genes such as hilA and invA. hilA is required for downstream gene transcription and epithelial cell invasion [4]; invA is responsible for invasion of Salmonella species into the host for initiating the infection. Therefore, the increasing degree of pathogenicity of the isolates and the amplification of nucleotide sequences within the hilA and invA of Salmonella have been evaluated as a means of detecting invasive Salmonella serovars [5].
Line 275 Figure 1F; Line 141-148. 2.6.
What is the house keeping gene used for normalization in the qRT-PCR for the mRNA expression levels of Salmonella virulence factors HILA and INV1? The unit of Y axis should be “relative expression level to *** gene (housekeeping gene)”.
Reply:
16S rRNA was used as the house keeping gene for ormalization in the qRT-PCR. We also revised the unit of Y axis.
Comment: Line 163
ormalization is a typing error. It should be normalization.
Line 325 Figure 3.
Pathways of neurodegeneration – multiple diseases, Huntington disease, Alzheimer disease showed associated transcriptional different changes during L. reuteri inhibits S. enterica process. Some other pathways such as hepatocellular carcinoma, Kaposi sarcoma-associated herpesvirus infection, EB virus infection, HPV infection are associated. What are the reasonable explanations for these associations?
Reply:
The reasons for the enrichment of pathways related to neurodegenerative diseases have been described above, while other signaling pathways such as Epstein-Barr virus infection are enriched in cluster 1 with a p.adjust > 0.05, which is not statistically significant. Some other signaling pathways associated with viral infections such as Kaposi sarcoma - associated herpesvirus infection (https://www.kegg.jp/entry/map05167), EB virus get infection (https://www.kegg.jp/entry/map05169), HPV infection (https://www.kegg.jp/entry/map05165), many of the genes in these signaling pathways overlap with those in PI3K-Akt signaling pathway, MAPK signaling pathway, Cell cycle, Apoptosis and other pathways. This may also be an important reason why enrichment analysis can annotate pathways related to these viral infections.
Comment:
It is not understandable why the enrichment of pathways without statistical significance (e.g. EG virus infection) are shown in Figure 3, which becomes confusing to readers. Please focus on expressing the major results with statistical significance and give the major results correspondent discussion.
Line 425-427
L. reuteri ATCC 53608 restores the S. enterica BNCC186354-induced signaling changes in phosphorylated signaling molecules except those related to the cell cycle.
Reply:
Done.
Comment:
Line 448-449, Figure 7C
How can be the meaning of the revised sentence inn Line 448-449 shown in Figure 7C? In the revised Figure 7C, no impact upon cell cycle cannot be shown. In other words, L. reuteri inhibits S. enterica outside the IPEC-J2 cells, but only cell cycle not affected by such an inhibitory effect? Please try to express this key point in the revised Figure 7C in which the three blue dotted lines within the IPEC-J2 cell had been deleted.
Round 3
Reviewer 2 Report
The authors showed their endeavour in the revision.
Please revise all "salmonella" into "Salmonella" thorought the entire text, including lines 23, 347, 351, 500, and any others.
"Salmonella" on line 351 should be typed in Italic as "Salmonella".